# Can We Leave Deepfake Data Behind in Training Deepfake Detector?

**Jikang Cheng**[1][*] **Zhiyuan Yan**[2]**, Ying Zhang**[3]**, Yuhao Luo**[2]**, Zhongyuan Wang**[1][†]**, Chen Li**[3]
[1] School of Computer Science, Wuhan University
[2] The Chinese University of Hong Kong, Shenzhen (CUHK-Shenzhen)
[3] WeChat, Tencent Inc.
`ChengJikang@whu.edu.cn`, `wzy_hope@163.com`

## Abstract

The generalization ability of deepfake detectors is vital for their applications in real-world scenarios. One effective solution to enhance this ability is to train the models with manually-blended data, which we termed "blendfake", encouraging models to learn generic forgery artifacts like blending boundary. Interestingly, current SoTA methods utilize blendfake *without* incorporating any deepfake data in their training process. This is likely because previous empirical observations suggest that vanilla hybrid training (VHT), which combines deepfake and blendfake data, results in inferior performance to methods using only blendfake data (so-called "1+1<2"). Therefore, a critical question arises: Can we leave deepfake behind and rely solely on blendfake data to train an effective deepfake detector? Intuitively, as deepfakes also contain additional informative forgery clues (*e.g.,* deep generative artifacts), excluding all deepfake data in training deepfake detectors seems counter-intuitive. In this paper, we rethink the role of blendfake in detecting deepfakes and formulate the process from "real to blendfake to deepfake" to be a *progressive transition*. Specifically, blendfake and deepfake can be explicitly delineated as the oriented pivot anchors between "real-to-fake" transitions. The accumulation of forgery information should be oriented and progressively increasing during this transition process. To this end, we propose an Oriented Progressive Regularizor (OPR) to establish the constraints that compel the distribution of anchors to be discretely arranged. Furthermore, we introduce feature bridging to facilitate the smooth transition between adjacent anchors. Extensive experiments confirm that our design allows leveraging forgery information from both blendfake and deepfake effectively and comprehensively. Code is available at *https://github.com/beautyremain/ProDet*.

## 1 Introduction

In recent years, the development of deepfake[3] has aroused significant concerns regarding privacy and security among the public. Deepfake detection aims to identify whether a face from an unknown source has been manipulated by deepfake techniques. Most detection methods perform promisingly when trained and tested on identical manipulations. However, given the unpredictability and complexity of real-world scenarios, it is increasingly critical for these methods to generalize and identify previously unseen manipulations. One particularly effective and mainstream solution is data synthesis, which involves image blending to create new training forgery samples [7, 28, 39, 27, 55].

---

[*]Work done during an internship at WeChat
[†]Corresponding author.
[3]The term "deepfake" hereafter specifically refers to **face** forgery technology. The entire (or natural) image synthesis is not within our scope.

38th Conference on Neural Information Processing Systems (NeurIPS 2024).

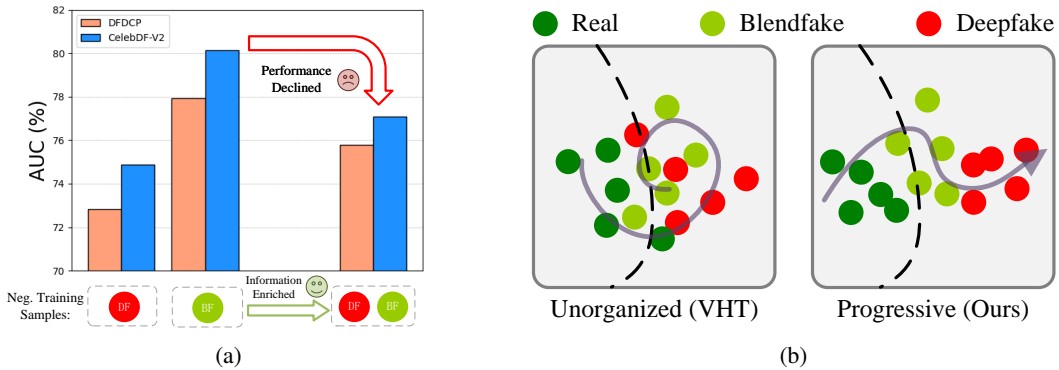

(a)  (b)

Figure 1: 1a: The detection performance experiences an abnormal decline when naively combining deepfake and blendfake as the negative sample for training, even though the forgery information is enriched in this process. 1b: Illustration Example for illustrating latent space organization. With progressively organized latent space (ours), information in both deepfake and blendfake is effectively leveraged, and deepfake samples become easier to distinguish from the real. See Fig. 4a and 9 for experimental results of actual latent-space distribution.

Specifically, these methods aim to synthesize *pseudo-fake* data that share forgery similarities with the actual deepfake data. The pseudo-fake data is generated by blending a real face with a near-landmark fake face [7], near-landmark real face [55, 27], or a real face itself [28, 39]. Considering the "deep" process is not involved during pseudo-fake data generation (*without* utilizing any deep network), we term these manually-blended pseudo-fake faces as *blendfake* faces to differentiate them from *deepfake* faces. The learned information from training on blendfake is similar to that on Deepfake: by simulating the blending operations in the Deepfake generation process, detectors trained on blendfake can identify generalized blending clues. In comparison to deepfake, blendfake necessitates the use of two images with closely matching landmarks (or with itself) for generation, thereby curtailing its practical utility. However, this also means that blendfake contains fewer forgery clues compared with Deepfake and its distribution is closer to that of real images, making blendfake a conveniently obtained hard sample for detector training.

In this paper, we commence with a question that many researchers may ponder: *Can the blendfake face entirely substitute the actual AI-generated deepfake face in training deepfake detectors?* As shown in Fig. 1a, considering the empirically inferior performance of using both deepfake and blendfake for training, exclusively utilizing blendfake as the negative sample becomes the mainstream scheme for data-synthesis methods [39, 27], where actual deepfake data is deliberately excluded. However, given that actual deepfake data is widely available in real-world scenarios, and it should contain extra forgery clues (*e.g.*, generative artifacts in deep net) absent in blendfake, training a deepfake detector without using any deepfake data seems counter-intuitive. Therefore, we argue that the significance of deepfake samples is underestimated due to insufficient exploration of how to appropriately utilize these samples. As illustrated in Fig. 1b, we attribute the abnormal efficiency of VHT to the unorganized latent-space distribution, while a carefully organized latent-space distribution is proven advantageous to network performance [26, 53, 2]. Specifically, since deepfake and blendfake are entangled while also distinctly containing forgery clues, the direct VHT may fail to disentangle the learned representation in the latent space. Inspired by introducing inductive bias that can reflect the structure of the underlying data (*i.e.*, real, blendfake, and deepfake in our case) is beneficial to feature robustness and generalization ability [19, 13, 23], we want to organize the latent space based on an inductive observation. As shown in Fig. 2, it can be observed that the transition from real to fake is a progressive process. Based on the observation, we posit that the latent space distribution from real to blendfake to deepfake should also be conceptualized as a *progressive process*, where real can gradually become deepfake through a continuous transition. That is, in the latent feature space, the distribution of blendfake and deepfake can be regarded as "oriented pivot anchors" in the progressive transition from real to fake. Achieving this progressive organization can disentangle learned representations of blendfake and deepfake data, thereby enabling more effective use of forgery clues from both data types. Consequently, we propose ProDet (deepfake Detection with Progressive transition) to achieve the progressive transition in the latent space that we envisioned.

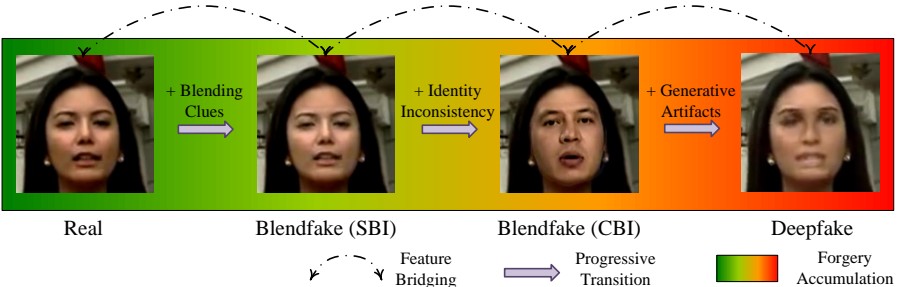

Figure 2: The progressive transition from real to fake, where blendfake and deepfake are explicitly delineated as the oriented pivot anchors according to their inherent forgery attributes.

As shown in Fig. 2, the oriented anchors used in this paper include four types: real, SBI [39], CBI [27], and fake, where both SBI (self-blended image) and CBI (cross-blended image) represent blendfakes. Within our framework, the anticipated progressive transition can be concretely understood as a progressive accumulation of forgery information, encompassing three specific aspects: the presence of *blending clues*, *identity inconsistency*, and *generative artifacts* of deep neural nets. To implement the aforementioned progressive effect, we consider ordering the distribution of these transition anchors by allocating different hard labels. Therefore, we introduce a module named Oriented Progressive Regularizer (OPR) to fully utilize the forgery information in all oriented anchors. OPR contains a triplet one-hot classifier corresponding to the forgery attributes and a projector to re-dimension the output auxiliary attention map, which is aligned with the feature dimensions and serves to guide the final detection results. Based on the inherent forgery attributes of different anchors, the label of the triplet one-hot output can be viewed as a cumulative three-bit binary number. Consequently, we establish the constraints that compel the distribution of anchors to be arranged in a progressive manner. To "build bridges" between the distinct adjacent anchors, we propose feature bridging to simulate continuous transitions between adjacent anchors. Specifically, feature bridging conducts the mixup to the extracted features of two adjacent anchors at random ratios, predicting mixed labels for these blended features to facilitate a smooth and continuous transition process. Additionally, to enhance the progressive transition process between different anchor distributions, we further propose the transition loss, which constrains an extracted feature to transform into its adjacent distribution after undergoing noise addition and specific mapping.

## 2 Related Works

### 2.1 Deepfake Detection Toward Generalization Ability

Deepfake detection aims to determine the authenticity of images potentially manipulated by deepfake approaches. Various detection methods concentrate on distinct facial features, such as movements of the lips [22] and the consistency of facial action units [4]. In parallel, several studies aim to optimize neural network architectures, including MesoNet [1], Xception [12], and CapsuleNet [34], to elevate detection efficacy. Based on the observation of model bias in the detector, many methods [51, 30] are proposed to remove certain general biases found in the forgery samples. Moreover, Chen *et al.* [20] construct a test sample-specific auxiliary task to improve the detection performance. To capture local-to-global information, Guan *et al.* [8] propose LTTD to focus on the temporal information within the local sequence. In the spectral domain, techniques like SPSL [31] and SRM [33] utilize phase spectra and high-frequency disturbances to augment the forgery details used in training. These strategies [31, 20, 8, 33, 22, 4, 24, 30, 51, 10, 6] target particular weaknesses found in deepfake technologies, and are highly effective in identifying deepfake images characterized by these flaws.

### 2.2 Deepfake Detectors with Blendfake Faces

Blendfake refers to a series of forgery detection methods that employ data generation schemes. These methods [7, 28, 39, 27, 55] manually create new fake samples that imitate general forgery clues that are similar to deepfake based on real samples from the dataset. Specifically, SLADD [7] dynamically crops a portion of a fake image and blends it onto a real image, thereby generating a fake sample that is more challenging to distinguish. Both CBI [27] and I2G [55] involve randomly selecting a real

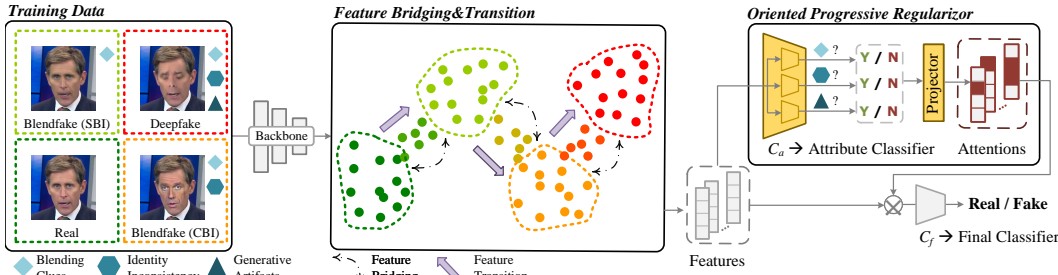

Figure 3: Overall pipeline of our method.

image as the base image, and then searching for another source real image with closely matching facial landmarks. Following a color transfer, the face of the source real image is attached to the base image according to the landmarks, ultimately producing a new fake sample. The resulting blendfake image is cross-face blended and exhibits inconsistencies in identity and also blending clues. FWA [28] and SBI [39] omit the landmark matching process, instead generating two different versions of a base real image through distinct transformations, and subsequently blending these two versions to create a new fake image. The blendfake images generated in this way contain only blending clues, causing their distribution to resemble real images more closely.

Owing to their superior implementation designs, SBI and CBI have demonstrated advanced performance in deepfake detection. Notably, they chose to use the generated blendfake solely instead of deepfake during training, thereby completely excluding deepfake from the detector training process.

## 3 Methodology

In this paper, we aim to devise an appropriate scheme to simultaneously leverage the forgery information inherent in both deepfake and blendfake, thereby training a more effective detector termed ProDet. Firstly, we consider constraining the oriented anchors, that is, the real, blendfake, and deepfake features, to be progressively distributed in a discrete manner. Then, we propose feature bridging to simulate the continuous and progressive transition between adjacent anchors. Lastly, a transition loss is introduced further to enhance our simulation of the progressive transition process. As shown in Fig. 3, our method is simple yet effective in organizing the latent space in a progressive manner.

### 3.1 Anchoring Oriented Distributions in Latent Space

To regularize a progressive distribution in latent space, we design a module termed Oriented Progressive Regularizor (OPR) using multi-attribute classification constraints. Specifically, blendfake images generated by both SBI [39] and CBI [27] are introduced as oriented pivot anchors to construct the process of progressive transition. Since SBI contains no clues of ID inconsistency compared to CBI, it is notably more difficult to identify SBI as fake data. Consequently, in a mini-batch during training, we incorporate four types of samples aligned with the same frame in the same video. They can be organized progressively from real to fake as follows: real ($\mathbf{I}_r$), SBI ($\mathbf{I}_s$), CBI ($\mathbf{I}_c$), and deepfake ($\mathbf{I}_d$), as depicted in Fig. 2. To facilitate progressive anchor organization, we employ an attribute classifier $\mathcal{C}_a$ that introduces three two-class classification heads. Instead of independently discerning the oriented anchors, where a lack of progressive relation is evident, three classification heads are tasked with the actual determination of *forgery attributes*, allowing for a progressive forgery accumulation. The classified two-class forgery attributes $\mathbf{A} = \{a_0, a_1, a_2\} \in [0, 1]^3$ respectively represent the likelihood of containing 1) Blending clue in $\mathbf{I}_s$, $\mathbf{I}_c$, and $\mathbf{I}_d$; 2) Identity inconsistency in $\mathbf{I}_c$, and $\mathbf{I}_d$; 3) Generative artifacts in $\mathbf{I}_d$. As illustrated in Fig. 3, it is straightforward to assign the ground-truth attribute label groups $\mathbf{A}' = \{a_0', a_1', a_2'\} \in \{0, 1\}^3$ to these oriented anchors, thereby enabling the progressive forgery accumulation through multi-labeling. Therefore, each image $\mathbf{I} \in \mathbb{R}^{H \times W \times 3}$ will be first extracted a feature $\mathbf{F} \in \mathbb{R}^{h \times w \times c}$ by the backbone encoder, then predicted with three forgery attributes:

$$\mathbf{A} = \mathcal{C}_a(\mathbf{F}). \tag{1}$$

To effectively leverage information from OPR during inference and obtain a conclusive deepfake detection result, the outputs of multi-attribute classification are re-dimensioned to an auxiliary

attention map $\mathbf{M} \in \mathbb{R}^{h \times w \times c}$ as:

$$\mathbf{M} = \mathcal{P}(\mathbf{A}), \tag{2}$$

where $\mathcal{P}$ is a projector that re-dimensions the classification results. $\mathbf{M}$ is aligned and integrated with the extracted feature $\mathbf{F}$ and provides supplemental information that improves the final deepfake detection outcome $y$. Therefore, $y$ can be written as:

$$y = \mathcal{C}_f(\mathbf{F} \odot \mathbf{M}), \tag{3}$$

where $\odot$ denotes element-wise multiplication, $\mathcal{C}_f$ denotes the final classifier for deepfake detection.

In fact, several alternative strategies can also achieve a similar multi-attribute classification for distribution anchoring, including multi-class, multi-label, and triplet binary (ours) classification. The multi-class classification is actually detecting different types of negative data instead of perceiving their forgery accumulation. Therefore, it cannot organize the latent space in a progressive manner as we anticipate. The multi-label strategy with three classes can be interpreted as using one vector to represent three parallel forgery attributes, which may achieve a comparable progressive accumulation. However, it is constrained by the mapping capability of a single detection head, limiting its effectiveness in distinguishing among different forgery attributes. In contrast, the deployed triplet binary strategy can effectively differentiate the distributions of the oriented anchor with progressive accumulation. The experimental results can be found in Sec. 4.3.

### 3.2  Simulating Continuous Transition via Feature Bridging

By assigning progressively multi-labels to different data distributions, we successfully placed discrete anchors for the real-to-fake progressive transition in the latent space. However, as analyzed in Sec. 1, the progressive transition from real to fake should be continuous rather than discrete. Therefore, we consider simulating this continuous progressive transition by constructing bridges between adjacent anchors, thus further clarifying the orientation. Here, we propose feature bridging to fill the continuous latent space between oriented anchors. Specifically, given four extracted features $\{\mathbf{F}_r, \mathbf{F}_s, \mathbf{F}_c, \mathbf{F}_d\}$ and their one-hot label groups $\{\mathbf{A}_r', \mathbf{A}_s', \mathbf{A}_c', \mathbf{A}_d'\}$ in an aligned mini-batch, we conduct mixup to each pair of adjacent features and labels. We take bridging real to SBI as an example:

$$\mathbf{F}_{r,s}, \mathbf{A}_{r,s}' = Mix(\mathbf{F}_r, \mathbf{F}_s), Mix(\mathbf{A}_r', \mathbf{A}_s'), \tag{4}$$

where $Mix(a, b) = \alpha \times a + (1 - \alpha) \times b$, and $\alpha \in [0, 1]$ is a random mixing ratio. It is important to note that, to ensure that the information mixed during the feature bridging process is the desired deepfake-related context and to exclude interference from deepfake-unrelated content (*e.g.,* background, identity, and pose), the images used for mixing should be aligned to the same video frame. Therefore, we can assign the forgery attribute label of $\mathbf{F}_{r,s}$ as $\mathbf{A}_{r,s}'$ to simulate the continuous transition between two adjacent anchors. Since feature bridging across skip-neighbor anchors introduces short-cut transition paths, thereby disrupting the oriented progressive transition in the "real-SBI-CBI-deepfake" process, we only conduct feature bridging on two adjacent anchors instead of any arbitrary pair.

### 3.3  Loss Function

**Transition Loss.** Given that a real distribution can be progressively transitioned to a fake distribution, features augmented with noise should be mapped to a more fake distribution. Therefore, we take this noise-augmented transition as an additional constraint to further enhance the simulation of progressive transition. Specifically, we first randomly generate random noise $\mathbf{N}$ that has the same shape as $\mathbf{F}$. Subsequently, we introduce a group of convolutional layers to merge and transform $\mathbf{F}$ and $\mathbf{N}$, producing a new feature $\mathbf{F}' \in \mathbb{R}^{h \times w \times c}$. We named the above process as feature transition ($\mathcal{T}$). Since $\mathbf{F}' = \mathcal{T}(\mathbf{N}, \mathbf{F})$ is expected to be transitioned to a more fake anchor distribution, we introduce the following transition loss:

$$L_t = \sum_{i=0}^{2} ||\mathcal{T}(\mathbf{N}, \mathbf{F}_i) - \mathbf{F}_{i+1}||_2, \tag{5}$$

where $\mathbf{F}_i$ represents the $i$-th feature in the mini-batch $\{\mathbf{F}_r, \mathbf{F}_s, \mathbf{F}_c, \mathbf{F}_d\}$, thus $\mathbf{F}_{i+1}$ denotes the subsequent feature to $\mathbf{F}_i$.

**Oriented Loss.** Oriented loss is deployed to both discrete anchors and continuous bridging features. Given a feature $\mathbf{F}$ and its one-hot label groups $\mathbf{A}' = \{a_0', a_1', a_2'\}$ either directly extracted or mixed,

its three attributes $\mathbf{A} = \{a_0, a_1, a_2\}$ predicted by $\mathcal{C}_t$ are constrained respectively. Formally, the oriented loss can be written as:

$$L_o = -\mathbf{A}' \log(\mathbf{A}^\mathrm{T}) - (1 - \mathbf{A}') \log(1 - \mathbf{A}^\mathrm{T}), \tag{6}$$

where $\mathbf{A}^\mathrm{T}$ denotes the transpose matrix of $\mathbf{A}$.

**Detection Loss.** We deploy detection loss on the discretely oriented anchors to constrain the final detection result used in both training and inference, which is formulated as:

$$L_d = y' \log(y) + (1 - y') \log(1 - y), \tag{7}$$

where $y'$ is the label that takes both blendfake and deepfake as the negative samples.

Therefore, the overall loss function in our method is:

$$L_{overall} = L_d + \beta L_o + \gamma L_t, \tag{8}$$

where $\beta$ and $\gamma$ are trade-off parameters. The algorithm of training the proposed deepfake detection with progressive transition in the latent space (ProDet) is summarized in Algorithm 1.

---

**Algorithm 1:** Training ProDet

---

**Input:** Dataset: $\mathcal{S}$; Training epoch: $\mathcal{E}$.
Initialize $\theta$ with the pre-trained backbone;
**for** $e = 1$ *to* $\mathcal{E}$ **do**
  **for** *mini-batch of aligned images and label groups* $\mathbf{I} = \{\mathbf{I}_r, \mathbf{I}_s, \mathbf{I}_c, \mathbf{I}_d\}$,
  $\mathbf{y}' = \{y'_r, y'_s, y'_c, y'_d\}$, $\mathbf{A}' \sim \mathcal{S}$ **do**
    extract features for each data
    $\mathbf{F} = \{\mathbf{F}_r, \mathbf{F}_s, \mathbf{F}_c, \mathbf{F}_d\} = \mathrm{Backbone}(\mathbf{I})$
    conduct feature bridging (FB) between adjacent features
    $\hat{\mathbf{F}}, \hat{\mathbf{A}}' = \mathrm{FB}(\mathbf{F}, \mathbf{A}')$
    anchor distributions via oriented loss
    $L_o = \mathrm{BCELoss}(\mathcal{C}_a(\{\hat{\mathbf{F}}, \mathbf{F}\}), \{\hat{\mathbf{A}}', \mathbf{A}'\})$
    enhance transition simulation via transition loss
    $L_t = \sum_{i,j} ||\mathcal{T}_f(\mathbf{N}, \mathbf{F}_{i \in \{r,s,b\}}) - \mathbf{F}_{j \in \{s,b,d\}}||_2$
    compute detection loss based on the projected auxiliary attention maps
    $L_t = \mathrm{BCELoss}(\mathcal{C}_f(\mathbf{F} \odot \mathcal{P}(\mathcal{C}_a(\mathbf{F}))), \mathbf{y}')$
    compute overall loss
    $L_{overall} = L_t + \beta L_o + \gamma L_t$
    update $\theta$ for minimizing $L_{overall}$ via backpropagation
  **end**
**end**
**Output:** learned model parameter $\theta$.

---

# 4 Experiments

## 4.1 Experimental Setting

**Datasets.** To extensively explore the generalization ability of deepfake detectors, the most advanced and widely used deepfake datasets are applied in our experiments. FaceForensics++ (FF++) [37] is constructed by four forgery methods including Deepfakes (DF) [15], Face2Face (F2F) [44], FaceSwap (FS) [18], and NeuralTextures (NT) [43]. FF++ with High Quality (HQ) is employed as the training dataset for all experiments in our paper. The base images to generate blendfake images are also from FF++ (HQ) real. For cross-dataset evaluations, we introduce Celeb-DF-v1 (CDFv1) [29], Celeb-DF-v2 (CDFv2) [29], DeepFake Detection Challenge Preview (DFDCP) [16], and DeepFake Detection Challenge (DFDC) [16].

**Implementation Details.** For preprocessing and training, we strictly follow the official code and settings provided by DeepFakeBench [52] to ensure fair comparison. EfficientNetB4 [42] is employed as the backbone of our detector. The trade-off parameters are set to $\beta = 1$ and $\gamma = 10$. The Adam optimizer is used with a learning rate of 0.0002, epoch of 20, input size of $256 \times 256$, and batch size of 24. Feature Bridging is deployed after a warm-up phase of two epochs. *Frame-level* Area Under Curve (AUC) and Equal Error Rate (EER) [52] are applied as the evaluation metrics of experimental results. All experiments are conducted on two NVIDIA Tesla V100 GPUs.

Table 1: Cross-dataset evaluations (AUC) from FF++ [37] (in-dataset) to CDFv1 [29], CDFv2 [29], DFDCP [17] and DFDC [17] (cross-dataset). C-Avg. denotes the average value of cross-dataset results. The best results are highlighted in **bold**. Cross-dataset improvements compared with the previous best one are written in pink. See *Appendix* for comparisons with video-based methods.

| Method | Venue | FF++ | CDFv1 | CDFv2 | DFDCP | DFDC | C-Avg. |
|---|---|---|---|---|---|---|---|
| Xception [12] | CVPR'17 | 0.9637 | 0.7794 | 0.7365 | 0.7374 | 0.7077 | 0.7403 |
| Meso4 [1] | WIFS'18 | 0.6077 | 0.7358 | 0.6091 | 0.5994 | 0.5560 | 0.6251 |
| FWA [28] | CVPRW'18 | 0.8765 | 0.7897 | 0.6680 | 0.6375 | 0.6132 | 0.6771 |
| EfficientB4 [42] | ICML'19 | 0.9567 | 0.7909 | 0.7487 | 0.7283 | 0.6955 | 0.7408 |
| Capsule [34] | ICASSP'19 | 0.8421 | 0.7909 | 0.7472 | 0.6568 | 0.6465 | 0.7104 |
| CNN-Aug [46] | CVPR'20 | 0.8493 | 0.7420 | 0.7027 | 0.6170 | 0.6361 | 0.6745 |
| X-ray [27] | CVPR'20 | 0.9592 | 0.7093 | 0.6786 | 0.6942 | 0.6326 | 0.6787 |
| FFD [14] | CVPR'20 | 0.9624 | 0.7840 | 0.7435 | 0.7426 | 0.7029 | 0.7433 |
| F3Net [36] | ECCV'20 | 0.9635 | 0.7769 | 0.7352 | 0.7354 | 0.7021 | 0.7374 |
| SPSL [31] | CVPR'21 | 0.9610 | 0.8150 | 0.7650 | 0.7408 | 0.7040 | 0.7562 |
| SRM [33] | CVPR'21 | 0.9576 | 0.7926 | 0.7552 | 0.7408 | 0.6995 | 0.7470 |
| I2G-PCL [55] | ICCV'21 | 0.9312 | 0.7112 | 0.6992 | 0.7358 | 0.6555 | 0.7004 |
| CORE [35] | CVPRW'22 | 0.9638 | 0.7798 | 0.7428 | 0.7341 | 0.7049 | 0.7404 |
| Recce [6] | CVPR'22 | 0.9621 | 0.7677 | 0.7319 | 0.7419 | 0.7133 | 0.7387 |
| SLADD [7] | CVPR'22 | 0.9691 | 0.8015 | 0.7403 | 0.7531 | 0.7170 | 0.7530 |
| SBI [39] | CVPR'22 | 0.8176 | 0.8311 | 0.8015 | 0.7794 | 0.7139 | 0.7814 |
| IID [24] | CVPR'23 | **0.9743** | 0.7578 | 0.7687 | 0.7622 | 0.6951 | 0.7462 |
| UCF [51] | ICCV'23 | 0.9705 | 0.7793 | 0.7527 | 0.7594 | 0.7191 | 0.7526 |
| ProDet (Ours) | - | 0.9591 | **0.9094**
(↑ 9.42%) | **0.8448**
(↑ 5.40%) | **0.8116**
(↑ 4.13%) | **0.7240**
(↑ 0.68%) | **0.8225**
(↑ 5.26%) |

## 4.2 Overall Performance on Comprehensive Datasets

In Tab. 1, we provide extensive comparison results with existing *state-of-the-art* (SoTA) deepfake detectors based on DeepFakeBench [52], where all methods are trained on FF++ (HQ) and tested on other datasets. The 18 methods provided in the table are within the benchmark with the exact same experimental setting as our method. EfficientB4 can be treated as the baseline of only using deepfake data. X-ray (CBI) and SBI are the baselines of only using one specific type of blendfake data. Since we effectively incorporate SoTA generative methods [39, 27] and actual deepfake data, our method outperforms other detectors in all evaluated cross-data metrics.

## 4.3 Ablation Study

In Tab. 2, we provide the baseline results in the upper part. Specifically, Blendfake-only (BF-only) represents using only blendfake data (*i.e.*, CBI [27] and SBI [39]) to train the deepfake detector. Vanilla Hybrid Training (VHT) represents simply incorporates all three data (*i.e.*, Deepfake, CBI [27], and SBI [39]) into training. It can be observed that VHT is outperformed by BF-only in all cross-dataset metrics, which demonstrates the empirical basis of solely using blendfake data.

**Overall Ablation.** In the middle part of Tab. 2, we provide the results of removing each proposed component sequentially to evaluate their effectiveness. $L_t$ can enhance the simulation of the progressive transition and thus improve the overall detection performance. Meanwhile, Feature Bridging (FB) plays a crucial role in filling the continuous space between discrete anchors. In the case of w/o $L_o$, OPR cannot constrain the anchors to organize progressively and can be treated as an operation of feature self-attention, while feature bridging functions as a simple manifold mixup. The inferior performances of w/o $L_o$ demonstrate the effectiveness of progressively organizing latent space, rather than relying on simple tricks that might coincidentally improve performance.

**Effect of Classification Strategies.** In Sec. 3.1, we analyze the effect of alternative strategies for multi-attribute classification in OPR. Here, we conduct experiments to validate the performance of three different strategies, that is, Multi-class (M-C), Multi-label (M-L), and Triplet Binary (TB) strategies. The results in the lower part of Tab. 2 show that M-C performs better when in-dataset while cross-dataset performance is only marginally improved. This is because the disentanglement of

Table 2: Ablations for each network component (AUC↑ and EER↓). All variants are trained on FF++ (in-dataset) and evaluated on other datasets (cross-dataset). BF-only represents using only blendfake data as the negative samples. M-C, M-L, and TB denotes Multi-Class, Multi-Label, and Triplet Binary strategies, respectively.

| Variant | FF++ | | CDFv1 | | CDFv2 | | DFDCP | | C-Avg. | |
|---|---|---|---|---|---|---|---|---|---|---|
| | AUC | EER | AUC | EER | AUC | EER | AUC | EER | AUC | EER |
| BF-only | 0.8096 | 0.2811 | 0.8413 | 0.2171 | 0.8006 | 0.2804 | 0.7791 | 0.3019 | 0.8070 | 0.2665 |
| VHT | 0.9353 | 0.1435 | 0.8145 | 0.2603 | 0.7710 | 0.2768 | 0.7577 | 0.3026 | 0.7811 | 0.2799 |
| w/o $L_o$ | 0.9311 | 0.1493 | 0.8401 | 0.2281 | 0.7959 | 0.2705 | 0.7901 | 0.2737 | 0.8087 | 0.2574 |
| w/o FB | 0.9601 | 0.0816 | 0.8696 | 0.2001 | 0.8278 | 0.2537 | 0.8037 | 0.2811 | 0.8337 | 0.2449 |
| w/o $L_t$ | 0.9535 | 0.1326 | 0.8890 | 0.1799 | 0.8356 | 0.2301 | **0.8174** | 0.2636 | 0.8473 | 0.2245 |
| M-C | **0.9677** | **0.0835** | 0.8630 | 0.2108 | 0.8092 | 0.2739 | 0.7965 | 0.2658 | 0.8229 | 0.2501 |
| M-L | 0.9576 | 0.0994 | 0.8757 | 0.1893 | 0.8229 | 0.2533 | 0.7939 | 0.2748 | 0.8308 | 0.2391 |
| TB (**Ours**) | 0.9591 | 0.1014 | **0.9094** | **0.1688** | **0.8448** | 0.2136 | 0.8116 | **0.2628** | **0.8553** | **0.2151** |

Table 3: Ablations on leveraging oriented anchors progressively (AUC). All variants are trained on FF++ (in-dataset) and evaluated on other datasets (cross-dataset).

| SBI | SBI-FB | CBI | CBI-FB | DF | DF-FB | FF++ | CDFv1 | CDFv2 | DFDCP | C-Avg. |
|---|---|---|---|---|---|---|---|---|---|---|
| ✓ | | | | | | 0.8176 | 0.8311 | 0.8015 | 0.7794 | 0.8040 |
| ✓ | ✓ | | | | | 0.8343 | 0.8507 | 0.8136 | 0.7659 | 0.8101 |
| ✓ | ✓ | ✓ | | | | 0.8191 | 0.8439 | 0.7917 | 0.7910 | 0.8089 |
| ✓ | ✓ | ✓ | ✓ | | | 0.8210 | 0.8551 | 0.8151 | 0.8081 | 0.8254 |
| ✓ | ✓ | ✓ | ✓ | ✓ | | 0.9539 | 0.8891 | 0.8336 | 0.7947 | 0.8391 |
| ✓ | ✓ | ✓ | ✓ | ✓ | ✓ | **0.9591** | **0.9094** | **0.8448** | **0.8116** | **0.8553** |

one-hot values allows deepfake learning independently but does not utilize both datasets for cross-dataset generalization. M-L performs similarly to the proposed TB since it can achieve progressive transition. Still, our method exhibits superior effectiveness considering its improved capability in discerning different forgery attributes.

**Progressively Leveraging Data.** To further verify the superiority of the progressive transition distribution, we explore the effects of various hybrid training schemes from an experimental perspective. As shown in Tab. 3, we consider progressively incorporating pivot anchor data into the training samples from real to deepfake, and augmenting anchors with feature bridging (FB). Overall, by learning a progressive transition process, our method is capable of effectively utilizing forgery information contained within both blendfake and deepfake data.

## 4.4 Robustness against Unseen Perturbations

To comprehensively evaluate the robustness against unseen perturbations of our method, we conduct experiments from two perspectives, that is, feature robustness (mPD) and detection robustness (AUC). In Tab. 4, we provide the detailed performance on the three types of distinct unseen perturbations, that is, Block-wise masking (Block-wise), Gaussian noise (Noise), and Shifting (Shift). All perturbations are applied with random intensity ten times. Namely, Block-wise is applied with a ratio of 0.1 and $4 \times 4$ grids. Noise is applied with a random variance selected from the range of 10 to 50. Shift is applied with the range of -50 to 50 on both x and y axes. In Fig. 5, we further provide the robustness comparisons with multiple perturbation intensities. The superior robustness of our method stems from the organized latent space, which allows the network to leverage the rich information in hybrid training more effectively.

## 4.5 Analysis of Learned Feature in Latent Space

To obtain features with dimensions that can be directly visualized, we cleverly construct toy models to facilitate our analysis of the learned feature distribution. The reasons for not applying clustering visualization methods (*e.g.*, t-SNE [45]) are demonstrated in *Appendix*. Specifically, we modify the

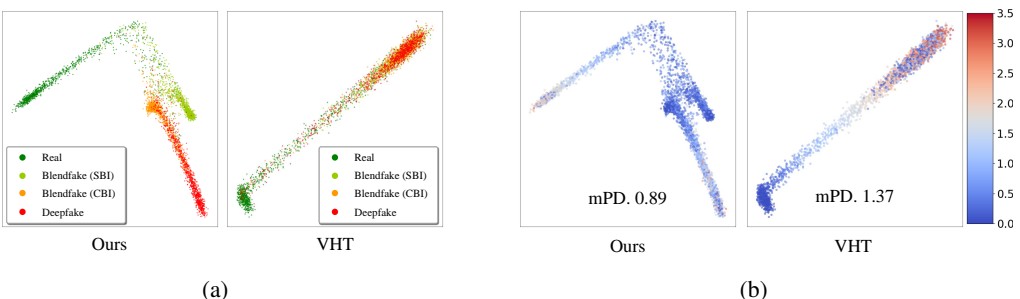

|       |       |
| :---: | :---: |
| Ours  | VHT   |
| (a)   |       |

| Ours  | VHT   |
| :---: | :---: |
| mPD. 0.89 | mPD. 1.37 |
| (b)   |       |

Figure 4: 4a: Illustration of feature organization, where our method can organize different anchors in a progressive manner, while VHT is unorganized and fails to discern blendfake and deepfake. 4b: Illustration of feature regularity. The heatmap values represent the PD at each point, and mPD is the mean PD in the distribution, while smaller mPD implies better feature regularity. The results show that our method has a smaller mPD and superior regularity.

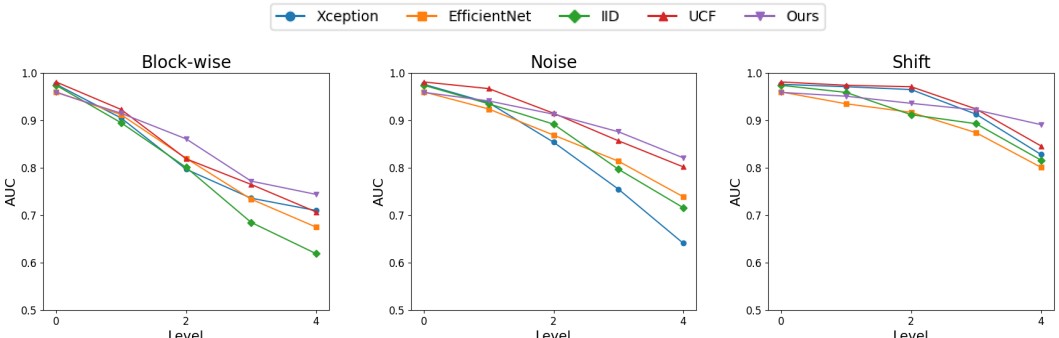

Figure 5: Robustness against unseen perturbation with different intensity levels.

architecture of the backbone to extract features from 512 to 2 dimensions. Subsequently, we train two toy models using VHT and our approach while keeping all other settings consistent. Based on these toy models, we can easily visualize the actual learned features directly in the 2D space.

**Feature Organization.** In Fig. 4a, it can be observed that the distribution of features extracted by our method is aligned with our anticipated progressive transition. Simultaneously, our method enhances the separation between real and fake features, leading to improved clustering and more effective detection. In contrast, the features from VHT are unorganized and highly entangled, which is detrimental to both decision-making and feature generalization ability.

**Feature Regularity.** Feature regularity measures the domain shifting issue in the latent space, while a better regularity implies that the learned features are more accurate, robust, and generalizable [41, 5, 54]. To analyze the regularity of the latent space organization achieved by our progressive transition scheme, we apply random unseen perturbations to each data point and calculate a new metric named Perturbed Distance (PD). A smaller PD suggests that the image with its perturbed versions is mapped more closely in the latent space, indicating superior feature regularity and smoothness. Subsequently, mPD denotes the mean PD in a distribution. See *Appendix* for the precise formulation and definition of PD and mPD. As shown in Fig. 4b, the result demonstrates that our scheme can organize the latent space with enhanced regularity, which is beneficial for generalization. It also indicates the improved robustness of our method against unseen perturbations.

### 4.6 Evaluation on Alternative Organized Distribution

To further establish the superiority of the proposed organization of latent space, we conduct comparisons with other feasible latent space distributions, including **1)** Real to Deepfake to Blendfake transition (R2D2B). **2)** Like a circle, blendfake and deepfake surround real with similar distances (Surround). The proposed Real to Blendfake to Deepfake transition is denoted by R2B2D. In Tab. 5, it can be observed that R2D2B performs the worst since it violates the inductive observation, that

Table 4: The mPD↓ and AUC↑ for robustness against unseen perturbations on FF++ test set.

| Method | Block-wise | | Noise | | Shift | | Avg. | |
|--------|--------|--------|--------|--------|--------|--------|--------|--------|
| | mPD | AUC | mPD | AUC | mPD | AUC | mPD | AUC |
| VHT | 1.8124 | 0.7978 | 1.4802 | 0.8345 | 0.8317 | 0.8737 | 1.3748 | 0.8353 |
| Ours | **0.9153** | **0.8694** | **1.0355** | **0.8431** | **0.7106** | **0.9015** | **0.8871** | **0.8713** |

Table 5: Evaluation on different latent space organizations (AUC). All variants are trained on FF++ (in-dataset) and evaluated on other datasets (cross-dataset).

| Distribution Organization | FF++ | CDFv1 | CDFv2 | DFDCP | C-Avg. |
|--------|--------|--------|--------|--------|--------|
| Unorganized (VHT) | 0.9353 | 0.8145 | 0.7710 | 0.7577 | 0.7930 |
| R2D2B | 0.9552 | 0.7935 | 0.8126 | 0.7736 | 0.8090 |
| Surround | **0.9631** | 0.8836 | 0.8297 | 0.7863 | 0.8332 |
| R2B2D (**Ours**) | 0.9591 | **0.9094** | **0.8448** | **0.8116** | **0.8553** |

is, the real-to-fake progressive transition. Surround may be treated as another improved distribution organization compared with VHT, while it is still empirically inferior to the proposed organization.

## 5 Conclusion and Discussions

In this paper, we commence with a question that many researchers may ponder: Can we leave deepfake data behind in training deepfake detectors? We argue that the abnormal data efficacy of vanilla hybrid training may stem from the poor latent space organization. Hence, we propose to formulate the process of "real to blendfake to deepfake" to be a progressive transition. By progressively organizing blendfake and deepfake as oriented anchors and bridging them to simulate the continuous transition, the proposed ProDet effectively utilizes both data and exhibits superior detection performance.

**Limitations and future work.** It is intuitive to observe that the current deepfake is easier to distinguish compared to blendfake. Naturally, we can organize them based on the "real-blendfake-deepfake" to improve the learned representation. However, with the advancement of deepfake techniques, all forgery attributes in deepfake data might be significantly reduced, thus making the intuitive argument that "deepfakes are easier to distinguish than blendfakes" potentially inaccurate. Nevertheless, to simultaneously utilize blendfake and deepfake data, a rational organization of the latent space still plays a crucial role. Therefore, in future work, we may identify more universally distinctive characteristics between blendfake and deepfake to effectively organize the latent space.

**Ethic Impacts.** The proposed method strives to minimize negative social impacts and provide stronger protection for privacy, including the societal harms introduced by AIGC and the implications and challenges posed by detection errors. Specifically, our method improves the effectiveness and generalization ability of the deepfake detector, which could be crucial for reducing errors and mitigating the negative impact of malicious deepfake content.

## 6 Acknowledgments and Disclosure of Funding

We would like to thank all the reviewers for their constructive comments. Our work was supported in National Natural Science Foundation of China (NSFC) under Grant No.62171324, No.62371350, and No.62072347.

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

# 7 Appendix

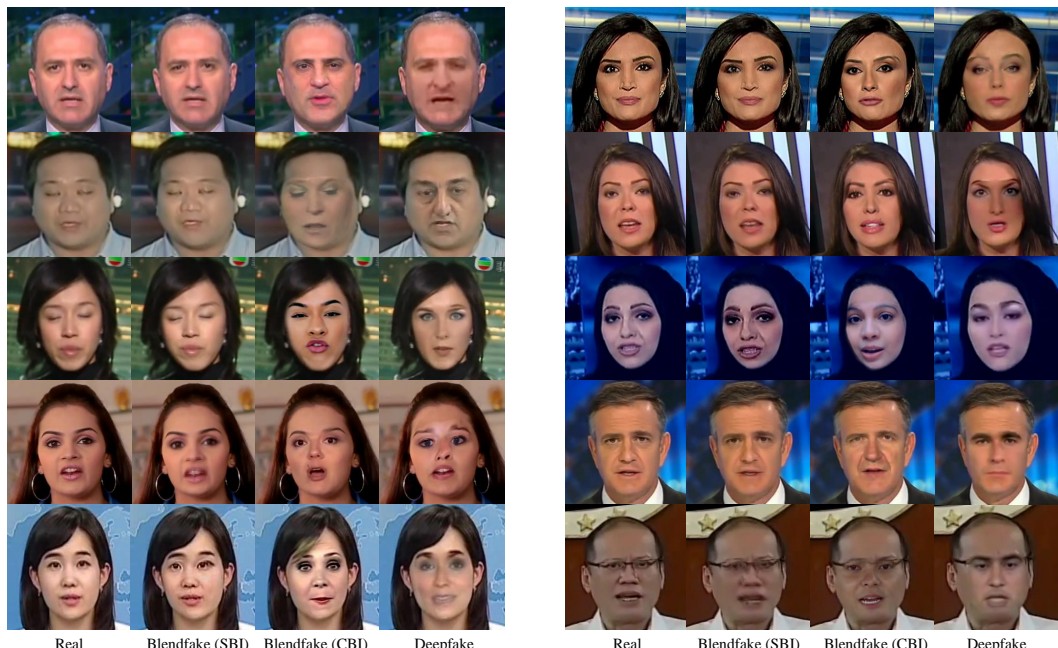

| Real | Blendfake (SBI) | Blendfake (CBI) | Deepfake | | Real | Blendfake (SBI) | Blendfake (CBI) | Deepfake |

Figure 6: The examples for the real-to-fake progressive transition.

## 7.1 More Perceptual Examples for the Inductive Observation of real-to-fake Progressive Transition

Considering that this paper is primarily based on the observation of a "progressive transition from real to fake", in Fig. 6, we present additional perceptual examples to provide a comprehensive impression. It is evident that sequentially from real to blendfake to deepfake, they become easier to distinguish as fake and the forgery information is accumulated. Therefore, it is intuitive to improve the latent space organization based on such inductive observation.

## 7.2 Detailed Experimental Environments for Reproducibility

In this paper, all experiments are conducted based on an open-source benchmark DeepfakeBench [52]. Specifically, for face extraction, we deploy an 81 facial landmarks shape predictor in Dlib [25] and select the detected rectangle with the biggest area in one frame. We use 32 frames in each video for both training and testing. During training, various data augmentation techniques are deployed including JPEG compression, random brightness and contrast, rotation, and median blur.

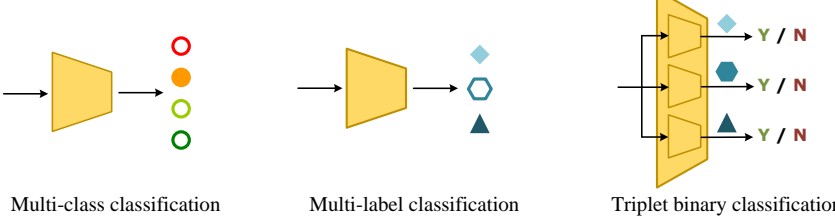

| Multi-class classification | Multi-label classification | Triplet binary classification |

Figure 7: Three different strategies for the multi-attribute classifier in OPR. The solid patterns signify the detected classes, whereas hollow patterns denote those that are not detected.

Table 6: Cross-dataset performance comparisons with video-based methods. We use *video-level* AUC as the evaluation metric. All methods are trained on FF++ [37] and test on other datasets.

| | Venues | Celeb-DF-v2 | DFDC | Avg. |
|---|---|---|---|---|
| LipForensics [22] | CVPR21 | 0.824 | 0.735 | 0.780 |
| FTCN [57] | ICCV21 | 0.869 | 0.740 | 0.805 |
| RealForensics [21] | CVPR22 | 0.857 | 0.759 | 0.808 |
| AltFreezing [47] | CVPR23 | 0.895 | - | - |
| Choi et al. [11] | CVPR24 | 0.890 | - | - |
| Ours | - | **0.925** | **0.770** | **0.848** |

## 7.3 Detailed Implementation of Classification Strategies

As shown in Fig. 7, we provide the detailed implementation of three different strategies. For the multi-class strategy with four classes, labels appear to be progressively assigned from 0 to 3. Namely, it can be treated as classifying real, SBI, CBI, and deepfake, respectively. However, the positions of each one-hot value are disentangled and can be rearranged arbitrarily. Therefore, the multi-class strategy is inherently unable to grant different anchors to be organized in a progressive manner. The latter two strategies can achieve similar effects of accumulating forgery attributes, therefore, their experimental performances are relatively closer. Still, we attribute the superior performance of triplet binary classification to its enhanced ability in disentangling forgery attributes.

## 7.4 Precise Formulation and Definition of Mean Perturbed Distance (mPD)

As we discussed in the main paper, we designed a new metric termed mean perturbed distance (mPD) to measure the feature robustness of each data point. Supposing the data point and its extracted feature as $\mathbf{I}$ and $\mathbf{F}$, applying perturbation on $\mathbf{I}$ can be written as $\mathcal{P}(\mathbf{I}, r)$, where $\mathcal{P}()$ is the perturbation function and $r$ is a randomly generated value for the perturbation intensity. By performing $\mathcal{P}(\mathbf{I}, r)$ with random $\mathcal{P}$ and $r$ and extracting their features, we can obtain a set of perturbed feature $\{\mathbf{F}_1, \mathbf{F}_2, ..., \mathbf{F}_n\}$. Then, we calculate the average Euclidean distances between the original features and ten perturbed features. Finally, we standardize the distances in distribution with different orders of magnitude. Therefore, the PD can be written as:

$$\text{PD} = \sum_{i=1}^{n} \frac{\sqrt{(\mathbf{F}_i)^2 + (\mathbf{F})^2}}{n\mathbf{F}_{std}},$$ 

(9)

where $\mathbf{F}_{std}$ denotes the standard deviation of each dimension in one distribution. Finally, the mPD of one distribution $\{\mathbf{F}^1, \mathbf{F}^2, ..., \mathbf{F}^m\}$ can be formulated as:

$$\text{mPD} = \sum_{j=1}^{m} \sum_{i=1}^{n} \frac{\sqrt{(\mathbf{F}_i^j)^2 + (\mathbf{F}^j)^2}}{mn\mathbf{F}_{std}}.$$ 

(10)

## 7.5 Generalization Assessment on Wider Scales

**Comparison with Video-level Methods.** Given that the proposed method makes decisions based on one single image frame, the main paper includes comparisons with other frame-level methods and measures their performance using *frame-level* AUC. Nevertheless, video-based deepfake detection methods can also achieve promising results. Therefore, in Tab. 6, we compare the cross-dataset generalization ability of our method with *state-of-the-art* video-based methods, taking *video-level* AUC as the metric. The results of comparison methods are copied from their official papers. By leveraging both blendfake and deepfake effectively, our method achieves the best performance among those methods.

**Evaluations on More Various and Advanced Datasets.** We further enlarge the evaluation scope by considering several key aspects. Firstly, we utilize nine different deepfake datasets to ensure the

Table 7: Generalization evaluations on comprehensive datasets.

| Methods | DFD | DF1.0 | FAVC | WDF | DiffSwap | UniFace | E4S | BlendFace | MobileSwap |
|---|---|---|---|---|---|---|---|---|---|
| DF-only | 0.8144/0.8621 | 0.7462/0.7474 | 0.8404/0.9150 | 0.7275/0.6883 | 0.7959/- | 0.7775/0.8212 | 0.6514/0.6955 | 0.7813/0.8296 | 0.8475/0.9053 |
| BF-only | 0.8378/0.8901 | 0.7345/0.7811 | 0.8627/0.9237 | 0.7563/**0.7965** | 0.8265/- | 0.6745/0.6998 | 0.6797/0.7113 | 0.8041/0.8529 | 0.8883/0.9399 |
| VHT | 0.8215/0.8505 | 0.7702/0.8312 | 0.8402/0.9125 | 0.7263/0.7811 | 0.7961/- | **0.8445**/0.8979 | 0.6704/0.7101 | 0.8311/0.8930 | 0.8729/0.9295 |
| Ours | **0.8581/0.9073** | **0.7902/0.8536** | **0.9077/0.9766** | **0.7718**/0.8287 | **0.8459**/- | 0.8441/**0.9077** | **0.7103/0.7711** | **0.8619/0.9287** | **0.9285/0.9748** |

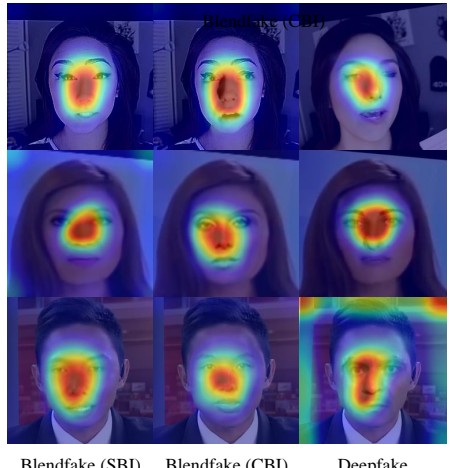 

| Blendfake (SBI) | Blendfake (CBI) | Deepfake | Blendfake (SBI) | Blendfake (CBI) | Deepfake |

Figure 8: Saliency map visualization for VHT (left) and our method (right).

diversity and comprehensiveness of the testing data in our evaluation. Secondly, we include the latest synthesis methods by using testing data from newly released deepfake datasets in 2024. These datasets incorporate advanced deepfake techniques such as UniFace [48], E4S [32], BlendFace [40], and MobileSwap [49] from DF40 [50], as well as DiffSwap [56] from DiffusionFace [9]. Finally, our evaluation design involves different variants, including Deepfake-only (DF-only), Blendfake-only (BF-only), VHT, and our proposed method. These variants are used in ablation studies using both *frame-level/video-level* AUC. As shown in Tab. 7, we can observe that our method consistently exhibits superiority in almost every testing data, which empirically suggests an improved generalization result.

## 7.6 Analysis of Attention Regions

To analyze the regions of interest in the network, we used Grad-CAM [38] to generate saliency maps for visualization. As shown in Fig. 8, VHT consistently focuses on partial regions of the face and fails to discern different fake data in aligned video frames. In contrast, our method can perceive forgery information from the broader regions of the face and the attention regions are more adaptive to different fake data.

## 7.7 Further experiments for learned feature representations.

### 7.7.1 Discussion on the t-SNE Visualization

Here, we demonstrate through experiments why we chose to construct toy models rather than directly visualizing the features extracted from original models using t-SNE [45]. In Fig. 9, the t-SNE and the actual feature distribution (see Fig. 4a) of the toy model are highly similar in terms of clustering. However, unlike the actual distribution, the progressive transitions between anchors are not precisely presented by t-SNE. This indicates that t-SNE is good at clustering [3] but performs limited in visualizing the distribution of progressive transitions, which is crucial for estimating our method. We also provide the t-SNE results of the original model. Based on our prior analysis of the t-SNE-feature pair results on the toy models, it can be anticipated that the progressive transitions in the t-SNE results

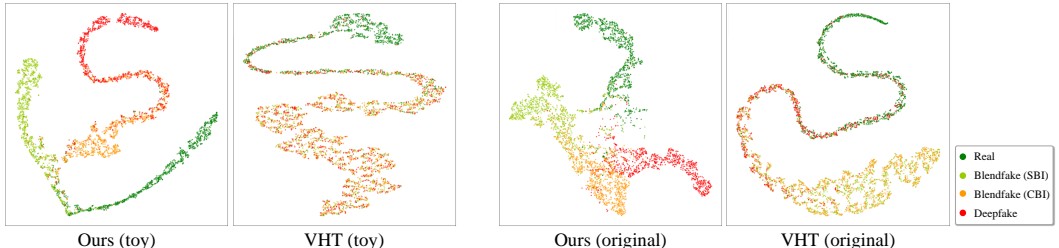

Figure 9: The t-SNE visualization for both toy and original models.

## DFDCP

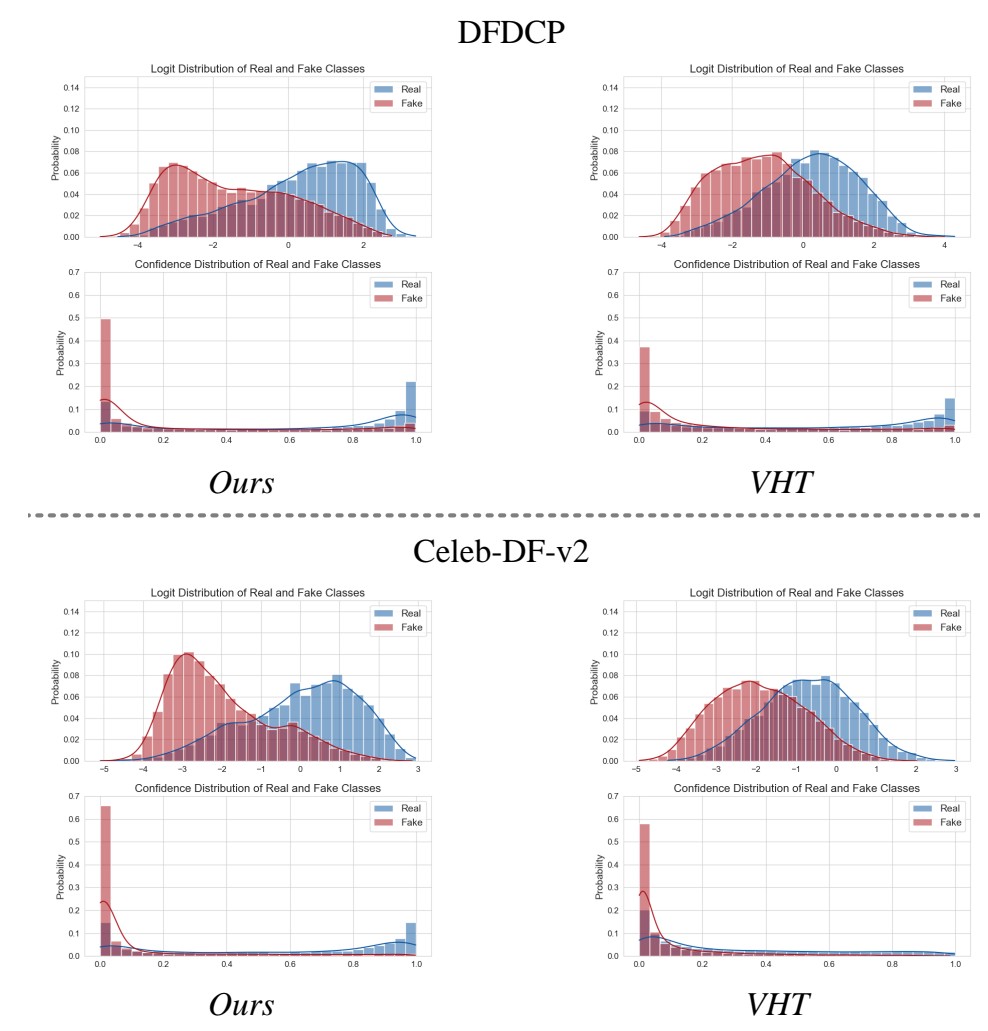

Figure 10: Distributions of the predicted confidence and logit outputs of *Ours* and *VHT* on DFDCP and Celeb-DF-v2. The y-value represents the probability of one sample falling within the corresponding x-value interval. Zoom in for better illustration.

will not be prominent. Nonetheless, it is still evident that our approach exhibits better clustering while exhibiting progressive transitions to a certain extent.

### 7.7.2 Distributions of the predicted confidence and logit outputs.

In Fig. 10, we provide a further investigation of the learned information of VHT and our method. Specifically, we summarize and analyze the distribution of logits output and confidence from VHT and Ours. We notice that VHT is less confident in both fake and real data. As we discussed, this

may be because naively combining DF and BF for training confuses the network, thus limiting its confidence in "understanding" the forgery representation of distinct BF and DF. In contrast, our model can predict both fake and real with high confidence since the model "understands" how real gradually becomes more and more fakes.

