# OpenReview forum: "Can We Leave Deepfake Data Behind in Training Deepfake Detector?"
_NeurIPS.cc/2024/Conference — NeurIPS 2024 poster_

### Official Review · Reviewer_6DHe · 2024-07-02

**Soundness:** 2
**Presentation:** 3
**Contribution:** 2
**Rating:** 4
**Confidence:** 5

**Summary:**

This study introduces a novel training strategy for Deepfake detection using real, blendfake, and deepfake datasets. By designing an oriented progressive regularizer and a feature bridging module, the proposed approach effectively extracts forgery information from the training data, resulting in enhanced generalizability.

**Strengths:**

The proposed method categorizes forgery faces into several types: SBI, CBI, and Deepfake faces, each containing distinct forgery artifacts, such as blending clues, identity inconsistencies, and generative fingerprints. The fine-grained learning scheme encourages the model to learn representative features from the training data, thus achieving robust and general face forgery detection.

**Weaknesses:**

1. The method employs a progressive transition from real to blendfake to deepfake samples. However, the necessity of continuity in these features remains unclear. The transition from real to fake faces, as depicted in Fig. 2, appears conceptually weird. The rationale behind the feature bridging and transition design is not well-explained. The progressive transition between adjacent anchors seems unusual, and the reasoning for a continuous rather than discrete transition is not justified.
2. Despite the generative artifacts present in deepfake data, it remains ambiguous why directly incorporating blendfake and deepfake data during training degrades performance. The authors suggest that direct VHT may fail to disentangle the learned representation in the latent space, but no experiments support this claim.
3. Fig. 1(b) does not appear to be an experimental result, which is crucial for validating the work's motivation.
4. In Line 44-45, the authors raised a question “Can the blendfake face entirely substitute the actual AI-generated deepfake face in training deepfake detectors?” However, this question has already been addressed by Face X-ray and SBI, which successfully use blendfake data to train general models.
5. The term A^T in Eq. (6) is not explained.
6. It is unclear why features augmented with noise should be mapped to a more fake distribution.
7. More Deepfake datasets, such as WildDeepfake and DeepForensics-1.0, should be included in cross-dataset evaluations.
8. For robustness evaluations in Table 6, the method should be compared with recent state-of-the-art deepfake detection methods, and more severity levels for each perturbation type should be included to mimic complex real-world scenarios.

**Questions:**

1. In Table 4, is R2B2D the same as D2B2R?
2. What is the real/fake label of the mix of F_r and F_s?

**Limitations:**

N.A.

---

> ### Author Rebuttal · Authors · 2024-08-04
>
> Thanks for the insightful comments. Here, we carefully clarify each issue mentioned by the respected reviewer.
>
> **Q1. What is the motivation for conceptualizing real to fake as a progressive transition?  Why it should be continuous rather than discrete?**
>
> **R1.** Thanks for this concern.  We endeavor to address this with a systematic and comprehensive response:
> - **Motivation of Progressive Transition**: Our paper seeks to effectively utilize both DF and BF data by taking the SBI and CBI as the intermediate anchors, where we **conceptualize "Real->SBI->CBI->DF" as a process of getting more and more fake**. It is intuitively apparent (Fig. 2 and Fig. 5 in the manuscript) and could be emphasized as accumulating forgery attributes (*i.e.*, blending clue, ID-inconsistency, and generative artifacts). Hence, we leverage this 'inductive bias' to organize the latent-space distribution, and thus achieve utilizing both DF and BF data effectively.
> - **Continuous or Discrete**: Building continuous bridges between adjacent anchors is based on the **inherent natures** and realizations of SBI, CBI, and DF. **Here, we would like to illustrate these natures in detail.**
>     - Taking Real->SBI as an example: SBI is generated by blending two versions of one Real transformed by augmentations (*e.g.*, blur, compression, and landmark shifting) with different severity levels. It can be emphasized that the unalignment levels of two transformed versions are controlled by an abstract parameter $\epsilon$, which represents the severity level of the blending clue. *Blending with a small $\epsilon$ produces an image that can be treated as Real with little perturbations, while large $\epsilon$ produces an SBI with severe blending clue*. Notably, we cannot assert a certain value of $\epsilon$ that when below it, the transformed image is fake and above it is real. That is, **we cannot define a clear decision boundary between Real and SBI**. Therefore, we can conclude that **the transition between Real and SBI is Continuous** according to the increase of blending clues.
>     - Analogously, SBI to CBI might controlled by the increase of ID-inconsistency, and from CBI to Deepfake might controlled by the generative artifacts. Both can be subsequently treated as continuous transitions.
>
> Therefore, we conceptualize the process of 'Real->SBI->CBI->DF' as a **progressive and continuous transition**. The necessity of continuity and progress, as we analyzed above, explains the rationale behind deploying transition and bridging. That is, the motivation to apply bridging and transition loss to simulate the conceptualized continuous and progressive transition.
>
> **Q2&3. Lacks experimental validation to the claimed motivation, that is, the latent space of VHT is not organized.**
>
> **R2&3.** Please refer to **Common Responses R2**, where we carefully address this concern.
>
> **Q4. The successful results in Face X-ray and SBI have already addressed the issue that we raised about substituting DF with BF.**
>
> **R4.** Thanks for your thoughtful consideration.  We would like to clarify it as follows.
> - As mentioned by the reviewer, the results of Face X-ray and SBI empirically show that **BF *may* substitute DF** considering 'DF-only < VHT< BF-only'. However, **it is precisely the 'stereotype' that our paper actually seeks to break down**.
> - The contribution and conclusion of our work is: using both of them to organize the latent space progressively can be more effective than using BF only, and thus **BF cannot substitute DF**. We show detailed analysis and validation for this point in **R2 of the Common Responses**.
> - This new answer is validated by the results of 'DF-only < VHT < BF-only < Ours' (Tab. 2 in manuscript).
>
> **Q5. $A^T$ is not defined**
>
> **R5.** Thanks for the kind mention. The superscript $T$ is commonly used to represent **transpose matrix**. We will explicitly define $T$ in the updated manuscript.
>
> **Q6. The rationale behind augmented with noise should be mapped to a more fake distribution**
>
> **R6.** We are grateful for your insightful concern. The transition loss is applied to **simulate the progressive transition between two adjacent anchors**. Specifically, we leverage '**noise+swallow network mapping**' to build a less complex mapping path between adjacent anchors, simulating the path of progressive transition. Therefore, augmenting noise to feature actually represents the transition relationship between two adjacent anchors. Whereas in the case of Real->BF->DF, it represents getting more fake.
>
> **Q7. More datasets (WDF and DF-1.0) should be included.**
>
> **R7.** We have provided more experimental results including the datasets you've mentioned. Please refer to **Common Responses R1**.
>
> **Q8. Robustness comparisons with SoTA and multi-intensity.**
>
> **R8.** Thanks. In Author-Rebuttal-PDF-Fig. 1, we hereby provide robustness evaluation with multi-level unseen perturbations and comparing with SoTAs. The robust performance of our method indicates its **effectiveness in complex real-world scenarios**. We will **update our manuscript** with this more comprehensive robustness evaluation.
>
>
> **Q9. In Table 4, is R2B2D the same as D2B2R?**
>
> **R9.** Thanks for this insightful concern. From the perspective of implementation, the code of R2B2D should be the same as D2B2R except for inverting the attribute labels. Therefore, **their experimental results are also expected to be consistent**. Theoretically, D2B2R should be considered as the opposite of R2B2D, that is, the conceptualized progressive transition is **getting more and more REAL**.
>
> **Q10. What is the real/fake label of the mix of F_r and F_s?**
>
> **R10.** Thanks for your concern. As illustrated **in Eq. 4 and line 172 of our manuscript**,  the label after bridging is assigned based on the mixing ratio $\alpha$. For example, the labels for $F_r$ is 0 and for $F_s$ is 1, and they are mixed with $\alpha=0.3$, the label for their mixed results should be $0.3×0+0.7×1=0.7$.

---

> ### Author Response · Authors · 2024-08-11
>
> Dear Reviewer 6DHe,
>
> Thank you for your thorough evaluation of our work. We are committed to incorporating your feedback comprehensively into our revised paper to improve both its content and overall quality.
>
> As the discussion period draws to a close, we hope our response has sufficiently addressed your concerns.   If there are any additional issues or points that require further clarification, we are more than willing to address them promptly.
>
> Best regards,
> The Authors

---

> > ### Comment · Reviewer_6DHe · 2024-08-13
> >
> > Thank you for the authors' response. Some of my minor concerns have been addressed. However, the justifications in responses R1-R4 are still not convincing. The proposed method shows limited robustness and has not been compared with the SOTA methods. Therefore, I will keep my rating unchanged.

---

### Official Review · Reviewer_Ssp1 · 2024-07-09

**Soundness:** 3
**Presentation:** 3
**Contribution:** 2
**Rating:** 5
**Confidence:** 5

**Summary:**

The authors introduced a method aimed at detecting deepfakes. Their approach, known as Oriented Progressive Regularizor (OPR), employs a progressive transition strategy. This strategy is designed to enable the model to effectively train on a combination of blendfake and deepfake data, ultimately leading to improved performance. The experimental results indicated that this method surpasses current state-of-the-art (SOTA) approaches when tested on deepfake datasets.

**Strengths:**

The paper provides a fresh perspective on the well-known problem of deepfake detection, which should be appreciated.

The paper is mostly well-written. The arguments and results presented are easy to understand.

The authors performed an extensive evaluation.

**Weaknesses:**

Argument on Blendfake: The argument that blendfake data alone is sufficient for training deepfake detectors is based on empirical observations on certain datasets or benchmarks with some particular deepfake detection models and may not hold universally. I would suggest toning down that claim or providing the exact conditions when this argument holds.

CDFv1 vs CDFv2: I believe that using CDFv1 for evaluation may not be necessary. It would have been more beneficial to utilize a different deepfake benchmark dataset such as FakeAVCele, DFD from Google/Jigsaw, or RWDF-23 (please refer to this repository for additional information https://github.com/Daisy-Zhang/Awesome-Deepfakes-Detection). The same applies to DFDC and DFDCP. I advocate for incorporating more diversity in the selection of benchmark datasets. In essence, the authors compared against three datasets instead of five, which is still an acceptable number.

Datasets: The authors utilized widely known deepfake datasets from 2019 in their research. However, considering the rapid advancements in deepfake technology since then, I believe these datasets may no longer accurately represent the current landscape. It would be valuable for the authors to include an assessment of their method using real-world deepfake videos sourced from social media and other online platforms. By doing so, they can demonstrate the effectiveness of their proposed solution in addressing contemporary and future iterations of deepfakes.

Progressive transition: Currently, the Progressive transition goes like this "Real --> Blendfake (SBI) --> Blendfake (CBI) --> Deepfake". I could imagine it being further extended to have addition of compression or adversarial artefacts (i.e., "Real --> Blendfake (SBI) --> Blendfake (CBI) --> Deepfake--> compression and other artefacts"). That way one could really see a generalisable pipeline that could incorporate the variance in the types of deepfakes available on social media and will greatly increase the quality of the work.

**Questions:**

See the above comments.

**Limitations:**

See the above comments.

---

> ### Author Rebuttal · Authors · 2024-08-04
>
> # **Response to Reviewer Ssp1**
> We are thankful for the reviewer's positive comments and interest in our research. We hope that the following point-by-point responses will enable the respected reviewer to further recognize our work.
>
> **Q1. The argument of 'blendfake data is sufficient' is based on empirical observations on certain datasets, which may be not the universe.**
>
> **R1.** We appreciate the reviewer's insightful observation. Our main point is that *while BF-only might not always outperform DF-only across all datasets, our method still demonstrates non-trivial advantages.*
> - To comprehensively investigate the effectiveness of DF and BF (using more datasets, not limited to "DF-series"), we enlarge our original evaluations by using **9** distinct deepfake datasets (see **Common Responses R1**). The results show that the situation of 'BF-only<DF-only' indeed sometimes exists, which validates the scoping concern from the respected reviewer.
> - However, **this does not undermine the significance of our method**. Namely, it can be observed that the VHT still fails to consistently achieve the effect of "1+1>2" and *shows degradation compared to BF-only*. In contrast, our method *effectively leverages both DF and BF data* and achieves superior performance compared with VHT, and so does DF-only and BF-only.
>
> **Q2. More advanced and in-the-wild datasets (FakeAVCele, DFD, and RWDF-23) are recommended to be included.**
>
> **R2.** We have provided more experimental results including the datasets you've mentioned. Please refer to **Common Responses R1**.
>
> **Q3. The progressive transition pipeline could be promoted to be more generalizable, for instance, by incorporating compression.**
>
> **R3.** We appreciate the reviewer's interest in gaining deeper insights into the proposed progressive transition pipeline. We value this suggestion very much and carefully conduct an experiment based on the pipeline of **'Real->Real (Compression)->SBI->CBI->DF'**. Notably, we do not place the Real (Compression) after DF as mentioned by the respected reviewer since we believe Compression is *not a faker version of DF*, considering it has no id-inconsistency or DF artifacts. Instead, Compression can be closer to the SBI, which actually involves compression operations in its creation process. The *frame-level* AUC results are shown below:
> |Methods|CDFv2|DFDCP|DFD|Avg.|
> |-|-|-|-|-|
> |Ours|0.8448|0.8116|0.8581|0.8382|
> |Ours+Compression|0.8351|0.8019|0.8604|0.8325|
>
> It can be observed that adding compression **exerts little effect** on experimental results, which may be due to the fact that random compression is already deployed as a data augmentation strategy for all training data. Still, we believe that this insight from the respected reviewer is **of great value**, which is very inspiring to us. For example, we may incorporate "**Facial Beautification Algorithm**" or "**Photoshop Image**" into the progressive pipeline to enhance the generalization and application scope. We are passionate about validating these ideas in our future works.

---

> ### Author Response · Authors · 2024-08-11
>
> Dear Reviewer Ssp1,
>
> We are encouraged by your thoughtful feedback and grateful for your interest in our research.
>
> As the discussion period coming to a close, we would like to inquire if our response has adequately addressed your concerns.  If there are any additional issues or points that require further clarification, we are more than willing to address them promptly.
>
> Best regards,
> The Authors

---

### Official Review · Reviewer_t2ao · 2024-07-12

**Soundness:** 2
**Presentation:** 3
**Contribution:** 2
**Rating:** 4
**Confidence:** 3

**Summary:**

This paper investigates the generalization ability of deepfake detectors and proposes a novel training approach using "blendfake" data to enhance the model's learning of generic forgery artifacts. The authors point out that existing state-of-the-art methods do not incorporate deepfake data in their training process, which contradicts previous empirical observations. The paper introduces an "Oriented Progressive Regularizor" (OPR) to establish constraints on anchor distribution and proposes feature bridging to facilitate smooth transitions. Experimental results indicate that the proposed method effectively utilizes forgery information from both blendfake and deepfake.

**Strengths:**

- Proposes a new training method that may enhance the generalization capability of deepfake detectors.
- Introduces OPR and feature bridging techniques to improve the model's recognition of forgery features.

**Weaknesses:**

- The attribution of the unorganized latent-space distribution lacks comprehensive experiments.
- There are some minor writing issues, such as the consistency of using SOTA and SoTA.

**Questions:**

- In ablation Table 2, the AUC of VHT is lower than that of BF-only in cross-dataset comparison, which is opposite to the statement in line 230.
- Are all the training sets for the SOTA model comparisons the same? How do you control the blendfake and deepfake training datasets for different models?

**Limitations:**

The authors believe that the reason VHT performs worse than blendfake-only is due to the unorganized latent-space distribution. Although the results indicate that the proposed method is effective, there is a lack of detailed experimental validation for attribution.

---

> ### Author Rebuttal · Authors · 2024-08-04
>
> # **Response to Reviewer t2ao**
> Thanks for your comments. Below we provide a point-by-point response to address the concern from the respected reviewer.
>
> **Q1. The attribution of the unorganized latent-space distribution lacks comprehensive experiments.**
>
> **R1.** Thanks for the concern. Please refer to **Common Responses R2**, where we comprehensively clarified this concern.
>
> **Q2. Writing issues**
>
> **R2.** Thanks. Please refer to  **Common Responses R3**.
>
> **Q3. Results in Tab. 3 may be opposite to our statement in line 230.**
>
> **R3.** Thanks for the concern. Actually, in line 230 we stated: 'VHT is **outperformed by** BF-only', which is **consistent** with the experimental results in Tab. 3, that is the reviewer mentioned 'VHT is **lower than** that of BF-only'.
>
> **Q4. Concerns about the training dataset for different methods**
>
> **R4.** Thanks for your thoughtful consideration. Here, we redesign the structure of the table to clarify our training data settings. All results are extracted and summarized from the corresponding parts in Tab. 1, 2, and 3 in our manuscript. For fair comparisons, all methods are deployed with the same backbone, *i.e.*, EfficientB4.
> | Methods| Training set | Celeb-DF-v1 |Celeb-DF-v2 | DFDCP |C-Avg.|
> |----------|----------|----------|----------|----------|----------|
> | EfficientB4  | DF  | 0.7909   | 0.7487  | 0.7283   | 0.7560 |
> | SBI  | SBI   | 0.8311  | 0.8015   | 0.7794   |  0.8040 |
> | Face X-ray  | DF,CBI | 0.7093   | 0.6786   | 0.6942   | 0.6940 |
> | BF-only  | SBI, CBI  | 0.8413   | 0.8006   | 0.7791   |  0.8070 |
> | VHT  | SBI, CBI, DF   | 0.8145   | 0.7710   | 0.7577   | 0.7810  |
> | Ours  | SBI, CBI, DF    | 0.9094   | 0.8448   | 0.8116  | 0.8553|
>
>
> Our paper holds the view that naively hybridizing more training data may undermine the cross-dataset performance (validated by BF-only > VHT). Therefore, the difference in training set **is not a biased setting**, instead, it demonstrates the superiority of our method in leveraging multiple data effectively.

---

> > ### Comment · Reviewer_t2ao · 2024-08-13
> >
> > I carefulky read through your respnose and apprecite your extra experiments. But the latend distribution concern remains, as evidence or experimental cues are missing, even with your response to R2. So I decidrd to hold my original decision.

---

> > > ### Author Response · Authors · 2024-08-13
> > >
> > > Dear Reviewer t2ao,
> > >
> > > We are grateful for your comments and feedback. We understand that **your biggest concern is about the evidence of latent distribution**.
> > >
> > > Actually, we have **already provided evidence of the latent distribution** from visualization approaches like **t-SNE.** Please see **Figure 4** and **Figure 6** in the manuscript. In these figures, we can see that the latent spaces of both VHT and Ours obey our expectation.
> > >
> > > If you have **any further questions** about our visualization and validation experiments, we sincerely anticipate further discussing and addressing any concerns you may have.
> > >
> > > Best, Authors

---

> ### Author Response · Authors · 2024-08-11
>
> Dear Reviewer t2ao,
>
> We deeply appreciate your dedicated efforts and insightful concern regarding our manuscript.
>
> With the discussion period coming to a close, we would like to inquire if our response has adequately addressed your concerns.  If there are any additional issues or points that require further clarification, we are more than willing to address them promptly.
>
> Best regards,
> The Authors

---

### Official Review · Reviewer_LWpa · 2024-07-22

**Soundness:** 3
**Presentation:** 4
**Contribution:** 3
**Rating:** 7
**Confidence:** 4

**Summary:**

The paper explores the utilization of blendfake and pseudo-fake data in training deepfake detectors. It argues that the significance of deepfake samples has been underestimated due to insufficient exploration. To better exploit both pseudo-fake and deepfake data, the paper introduces a progressive transition from "real to blendfake to deepfake" and proposes a hybrid training scheme. This scheme includes an oriented progressive regularizer (OPR) to model the transition and a feature bridging strategy to simulate a continuous transition.The paper explores the utilization of blendfake and pseudo-fake data in training deepfake detectors. It argues that the significance of deepfake samples has been underestimated due to insufficient exploration. To better exploit both pseudo-fake and deepfake data, the paper introduces a progressive transition from "real to blendfake to deepfake" and proposes a hybrid training scheme. This scheme includes an oriented progressive regularizer (OPR) to model the transition and a feature bridging strategy to simulate a continuous transition.

**Strengths:**

1.The paper is well-motivated, and the proposed solution is both intuitive and effective.
2.The experiments robustly demonstrate the rationality and effectiveness of the proposed design.

**Weaknesses:**

1. Choice of Blend Algorithms: The paper does not provide sufficient explanation or discussion on the choice of blendfake image algorithms (SBI and CBI). As mentioned in Section 2.2, there are many other methods for crafting blendfake images. Would these methods be effective as well?
2. Interpolation Strategy: In Section 3.2, the paper introduces an interpolation strategy to achieve a smoother transition from real to deepfake. Why was interpolation performed at the feature level, and would setting multiple mixing parameters (alpha) for more interpolations further improve performance?
3. Possible Typos: There might be a typo on line 149, $M_a$.

**Questions:**

See in weakness

**Limitations:**

See in weakness

---

> ### Author Rebuttal · Authors · 2024-08-04
>
> # **Response to Reviewer LWpa**
> We sincerely appreciate the reviewer's positive comments and rating on our paper, and the following are our point-to-point responses
>
> **Q1. Choice of Blend Algorithms： Why apply SBI and CBI instead of other Blendfake methods?**
>
> **R1.** Thanks for your thoughtful suggestion. Actually, the terms SBI and CBI in our paper **do not refer to two specific methods** (*i.e.*, SBI'22 [1] and X-ray), but a full set of methods with the **techniques of Self-Blended and Cross-Blended**. Therefore, we can surely take other Self-Blended methods or Cross-Blended methods in our current framework. To validate experimentally, we take FWA (another Self-Blended method) and I2G (another Cross-Blended method) to conduct a concise ablation study:
> | Methods | Blendfake Types | CDFv1 | CDFv2 | DFDCP | Avg. |
> |----------|----------|----------|----------|----------|----------|
> | VHT  | FWA, I2G   | 0.7894  | 0.6953   | 0.6575   | 0.7424   |
> | VHT  | SBI'22, X-ray   | 0.8145  | 0.7710   | 0.7577   | 0.7811   |
> | Ours   | FWA, I2G   | 0.8235   | 0.7746   | 0.7539   | 0.7990   |
> |Ours   | SBI'22, X-ray   |0.9094   | 0.8448   | 0.8116   | 0.8771   |
>
> It is evident that our method can also **enhance the performance of FWA&I2G** trained VHT model, while using SBI'22&X-ary performs essentially better.
>
>
> **Q2. The reason for applying interpolation on feature-level, and could mixing parameters $\alpha$ are set multiply for enhanced performance.**
>
> **R2.** We genuinely appreciate the consideration.
> - For the first question, the latent-space interpolation is performed to achieve **feature bridging** between two adjacent anchors (e.g., from SBI to CBI). This simulates the progressive and oriented transition of data becoming "more and more" fake starting from real. This process is illustrated intuitively in Figure 2 of the manuscript.
> - For the second question, $\alpha$ is randomly sampled from $U(0,1)$, where $U(a,b)$ denotes the Uniform distribution with the bounds of $a$ and $b$. Therefore, for diversity and enhanced performance, $\alpha$ is indeed a parameter with multiple values in our method.
>
> **Q3. Possible Typos**
>
> **R3.** Thanks. Please refer to **Common Responses R3**.
>
> [1] Kaede Shiohara and Toshihiko Yamasaki. Detecting deepfakes with self-blended images. In CVPR2022

---

> ### Author Response · Authors · 2024-08-11
>
> Dear Reviewer LWpa,
>
> We greatly appreciate your careful review and the insightful comments you shared regarding our manuscript.
>
> With the discussion period coming to a close, we would like to inquire if our response has adequately addressed your concerns.  If there are any additional issues or points that require further clarification, we are more than willing to address them promptly.
>
> Best regards,
> The Authors

---

### Author Rebuttal · Authors · 2024-08-04

We sincerely thank all reviewers for their valuable time and constructive comments, and we are strongly encouraged by their recognition of several strengths of our submission, including:
- **Fresh perspective/Well-motivated**  (Reviewers LWpa, Ssp1)
- **Extensive/Robust Evaluations** (Reviewers LWpa, Ssp1)
- **Well Presentation and Writing** (Reviewers LWpa, t2ao, Ssp1,6DHe)

Meanwhile, reviewers also indicate several important concerns and suggestions, which we will give **detailed common and individual responses**. Followings are the common responses.

# **Common Responses**
**Q1.** (**From Reviewers 6DHe, Ssp1**) **Cross-dataset evaluation is encouraged to include more datasets, such as {FakeAVCeleb, DFD, RWDF-23} (6DHe) and {WildDeepfake,  DeeperForensics-1.0} (Sspl).**

**R1.** Thanks for the valuable suggestion. Following reviewers' suggestions, we further enlarge the evaluation scope from the following aspects:
- **9 different deepfake datasets are used**: Ensuring the diversity and comprehension of the testing data in our evaluation;
- **Latest synthesis methods are considered**: Using the testing data from those **just-released deepfake datasets** (in 2024) with advanced deepfake techniques, that is, {UniFace, E4S,  BlendFace, MobileSwap} from DF40 [1] and {DiffSwap} from DiffusionFace [2];
- **Evaluation Design**: Involving different variants (i.e., Deepfake-only, Blendfake-only, VHT, and ours) for the ablation studies using both ***frame-level/video-level*** AUC.

Our results can be seen in the Table below.

| Methods| DFD |  DeeperForensics-1.0 | FakeAVCeleb | WildDeepfake| RWDF*| DiffSwap|UniFace| E4S|BlendFace | MobileSwap |
|----------|----------|----------|----------|----------|----------|----------|----------|----------|-----------|-----------|
| DF-only| 0.8144/0.8621| 0.7462/0.7474| 0.8404/0.9150|0.7275/0.6883| - |  0.7959/-  |0.7775/0.8212| 0.6514/0.6955   | 0.7813/0.8296  | 0.8475/0.9053|
| BF-only| 0.8378/0.8901  | 0.7345/0.7811   | 0.8627/0.9237|  0.7563/0.7965 |-| 0.8265/-| 0.6745/0.6998|0.6797/0.7113   | 0.8041/0.8529   | 0.8883/0.9399   |
| VHT   | 0.8215/0.8505   | 0.7702/0.8312| 0.8402/0.9125| 0.7263/0.7811   |  -| 0.7961/-| 0.8445/0.8979   | 0.6704/0.7101   |0.8311/0.8930   | 0.8729/0.9295   |
| Ours   | 0.8581/0.9073| 0.7902/0.8536   | 0.9077/0.9766| 0.7718/0.8287|  -  | 0.8459/ -| 0.8441/0.9077| 0.7103/0.7711|0.8619/0.9287| 0.9285/0.9748  |

**\*** *We are requesting the RWDF dataset (as suggested by Ssp1) from its authors. Once we have obtained the data, we will update the results immediately.*

From the Table, we can observe that our method consistently exhibits superiority in almost every testing data, which empirically suggests an improved generalization result.

**Q2.** (**From Reviewers 6DHe, t2ao**) **Why VHT (*naively combining deepfakes and blendfakes for training*) with *unorganized* latent space achieves *inferior* results than ours (with *organized* latent space).**

**R2.** We sincerely appreciate the insightful concern. We would like to address these concerns as follows:
- **Ambiguous Feature Space in VHT:** In VHT, deepfake (DF) and blendfake (BF) data are naively combined for training deepfake detectors. This approach results in an unorganized latent space where DF and BF data are intermixed, creating an ambiguous feature space.
- **Loss of Discriminative Features:** Due to the unorganized (mixed) latent space in VHT, the classifier tends to "regard" DF and BF as the same forgery type. This causes the classifier to "forget" the *discriminative characteristics of DF and BF*. However, DF and BF can be inherently different; DF contains more forgery information, such as generative artifacts produced by DNNs. Naively combing DF and BF for training (VHT) may result in the loss of these informative features, which can be crucial in detecting deepfakes.
- **Our Method's Advantages:**  Organizing latent space is a crucial topic that has been verified effective in many research domains [3,4]. In our case, the proposed method, which progressively organizes the latent space (real -> BF -> DF), can (1) help the model better 'understand' how real data gradually becomes fakes; (2) maintain more informative features to distinguish real from fakes. For these reasons, Ours should result in better generalization performance than VHT.
- **Experimental Validation:** For the claimed feature organization issue, we have verified this claim by visualizing the actual latent space (Fig. 4) and feature distribution (Fig. 6) in the manuscript. **Moreover**, in *Author-Rebuttal-PDF-Fig. 2*, we provide a further investigation of the learned information of VHT and our method. Specifically, we summarize and analyze the distribution of logits output and confidence from VHT and Ours. We notice that VHT is less confident in both fake and real data. As we discussed, this may be because naively combining DF and BF for training confuses the network, thus limiting its confidence in 'understanding' the forgery representation of distinct BF and DF. In contrast, our model can predict both fake and real with high confidence since the model "understands" how real gradually becomes more and more fakes.

**Q3.** (**From Reviewers LWpa and t2ao**) **Typos ($M_a$) and inconsistent spellings (SOTA and SoTA).**

**R3.** We are very grateful to the reviewers for their careful reading. We have carefully re-proofed our manuscript. We will continue to further optimize our writing and update our manuscript.

[1] Yan Z, Yao T, Chen S, et al. DF40: Toward Next-Generation Deepfake Detection. arXiv:2406.13495, 2024.

[2] Chen Z, Sun K, Zhou Z, et al. DiffusionFace: Towards a Comprehensive Dataset for Diffusion-Based Face Forgery Analysis. arXiv:2403.18471, 2024.

[3] Ali S and Kaick O. Evaluation of latent space learning with procedurally-generated datasets of shapes. ICCV 2021.

[4] Yang F and Ma C. Sparse and complete latent organization for geospatial semantic segmentation. CVPR 2022.

---

### Comment · Program_Chairs · 2024-08-14

Hi all, the author-reviewer discussion for this paper is extended to Aug 16 11:59pm ET for authors to reply to ethics reviews. -- PCs

---

### Decision · Program_Chairs · 2024-09-25

**Decision:**

Accept (poster)

**Comment:**

1x A, 1x BA, and 2x BR. This paper proposes a novel training strategy for Deepfake detection using real, blendfake, and deepfake datasets. The reviewers agree on the (1) clear writing, (2) fresh motivation, and (3) novel and effective method. Most of the concerns, such as the insufficient recent datasets, insufficient experiments on latent distribution, and insufficient robustness evaluation, have been addressed by the rebuttal. The ethical concerns on the proposed deepfake detector's limitations have also been cleared. Therefore, the AC leans to accept this submission.